# Outage-Based Resource Allocation for DF Two-Way Relay Networks with Energy Harvesting

**DOI:** 10.3390/s18113946

**Published:** 2018-11-14

**Authors:** Chunling Peng, Fangwei Li, Huaping Liu, Guozhong Wang

**Affiliations:** 1Chongqing Key Lab of Mobile Communications Technology, Chongqing University of Posts and Telecommunications, Chongqing 400065, China; chunlingp@163.com (C.P.); lifw@cqupt.edu.cn (F.L.); 2School of Electrical Engineering and Computer Science, Oregon State University Corvallis, OR 97331, USA; huaping.liu@oregonstate.edu; 3School of Communication Engineering, Chongqing College of Electronic Engineering, Chongqing 401331, China

**Keywords:** two-way relay, decode-and-forward, energy harvesting, joint resource allocation

## Abstract

A joint resource allocation algorithm to minimize the system outage probability is proposed for a decode-and-forward (DF) two-way relay network with simultaneous wireless information and power transfer (SWIPT) under a total power constraint. In this network, the two sources nodes exchange information with the help of a passive relay, which is assumed to help the two source nodes’ communication without consuming its own energy by exploiting an energy-harvesting protocol, the power splitting (PS) protocol. An optimization framework to jointly optimize power allocation (PA) at the source nodes and PS at the relay is developed. Since the formulated joint optimization problem is non-convex, the solution is developed in two steps. First, the conditionally optimal PS ratio at the relay node for a given PA ratio is explored; then, the closed-form of the optimal PA in the sense of minimizing the system outage probability with instantaneous channel state information (CSI) is derived. Analysis shows that the optimal design depends on the channel condition and the rate threshold. Simulation results are obtained to validate the analytical results. Comparison with three existing schemes shows that the proposed optimized scheme has the minimum system outage probability.

## 1. Introduction

Two-way relay (TWR) communications [1,2] to extend transmission range and improve communications reliability have been studied extensively. The traditional TWR network with two source nodes exchanging information between each other via a relay requires four time slots. Combined with appropriate network coding, the number of time slots can be reduced to two [3]. Two of the most commonly-used relay strategies are the amplify-and-forward (AF) [4] and DF [5] protocols. With the AF and DF protocols, plenty of significant research efforts designing optimal resource allocation schemes have been proposed, aiming at further improving essential performance objectives for such networks. For example, a power allocation (PA) scheme for an AF-TWR system and its achievable rates over Rayleigh fading channels were studied in [6]. In [7], based on maximizing the objective rate under a total power budget, a power allocation method for a specific DF relay network by exploiting physical-layer network coding was proposed. In [8], optimization of PA and time allocation (TA) was investigated to minimize the system outage probability to support asymmetric data rates for DF-TWR networks.

The aforementioned works all assumed that the relay spends its own power to support the TWR transmission. In most practical cases, it will be very attractive if the relay can harvest the energy it needs for relaying the information between the source nodes. One promising energy-harvesting technique is SWIPT, with which the relay performs energy harvesting (EH) by scavenging radio-frequency signals, typically transmitted by the source nodes, and then utilizes the harvested energy to forward information. Recent work has studied AF-TWR networks with SWIPT extensively. An AF-TWR with SWIPT that employs the PS protocol is proposed in [9] together with the derivation of tight upper- and lower-bounds of the outage probability and the ergodic capacity of the network. To further improve the AF-TWR performance, a joint optimization of the PS ratio and the transmission PA based on minimizing the outage probability was studied in [10]. A distributed energy beamforming scheme was proposed in [11] to maximize the achievable sum-rate of AF-TWR network with SWIPT. There are also some research efforts of SWIPT based on the DF-TWR network; for example, an energy-harvesting protocol for three-step TWR was proposed and analyzed [12]. A joint optimization scheme for a DF-TWR with energy harvesting was developed in [13].

In this paper, we focus on jointly optimizing the PS ratio and the PA ratio for a power-constrained, two-way relay network with a wireless powered relay that employs the DF protocol, a problem that has not been studied yet, to our knowledge. The major contributions of our works are summarized as follows. We form a jointly optimizing PS ratio and PA ratio problem to minimize the system outage probability in the context of a two-way energy-harvesting relay network that is subject to a total transmit power constraint. Since this original optimization problem is very difficult to solve, if not impossible, a two-step solution is developed to obtain a closed-form solution efficiently. Simulation results are obtained to verify the proposed algorithm and to assess the impact of various network parameters on system performance. The performances of three existing schemes are also simulated and compared with that of the proposed scheme.

The remainder of this paper is organized as follows. In Section 2, we describe the model of the two-way relay network being considered and formulate the joint resource allocation problem, aiming to minimize the outage probability. Transformation of the joint resource allocation problem to a two-step optimization problem is described in Section 3. Numerical results are presented in Section 4, and conclusions are given in Section 5.

## 2. System Model and Problem Formulation

### 2.1. System Model

The TWR network under consideration is shown in Figure 1, where two communicating devices $S_{1}$ and $S_{2}$ exchange their messages assisted by a passive relay node (no power is available for its information forwarding) with energy harvesting ability, denoted as *R*. It is assumed that there are no direct links between the two source nodes; thus, the information exchange must rely on the passive relay, which harvests the energy it needs from the RF signals transmitted by the two source nodes through the power splitting protocol. We assume all the nodes are single-antenna devices and operate in a half-duplex mode. In addition, we assume that the channel between any two nodes is reciprocal and that perfect channel state information is known at all nodes. The transmit power at $S_{1}$ and $S_{2}$ is denoted by $P_{1}$ and $P_{2}$, respectively, which share a total power budget constraint $P_{1}+P_{2}=P_{T}$ for each transmission round.

Two consecutive phases are involved to complete each round of information transmission: a multiple access (MA) phase and a broadcast (BC) phase.

During the MA phase, nodes $S_{1}$ and $S_{2}$ broadcast their signals simultaneously to relay *R*. The received signal at *R* in this phase is expressed as:(1)$$y_{R}=h_{1}\sqrt{P_{1}}x_{1}+h_{2}\sqrt{P_{2}}x_{2}+n_{a,w}$$where $h_{i}$ denotes the coefficient of the channel between $S_{i}$ and *R* and $n_{a,w}∼\;\mathrm{CN}\;\left(0,\sigma_{a}^{2}\right)$ is the noise generated at the receiver antenna.

The power splitter at *R* then splits the received signal $y_{R}$ into two portions ρ:(1−ρ), of which $\sqrt{\rho }y_{R}$ is utilized for energy harvesting, and the remaining portion in the amount of $\sqrt{1\;-\;\rho }y_{R}$ is used for information decoding. Let the energy conversion efficiency be η. The harvested energy at the relay is expressed as:(2)$$E=\frac{T}{2}\cdot \eta \rho (|h_{1}{|}^{2}P_{1}+|h_{2}{|}^{2}P_{2}+\sigma_{a}^{2})$$

Note that power splitting is done before the received signal is converted from passband to baseband; hence, the signal in the information decoding (ID) receiver can be expressed as:(3)$$\begin{array}{c} y_{\mathrm{ID}}=\sqrt{1\;-\;\rho }\left(h_{1}\sqrt{P_{1}}x_{1}\;+\;h_{2}\sqrt{P_{2}}x_{1}\;+\;n_{a,w}\right)\;+\;n_{b,w} \end{array}$$where $n_{b,w}\;∼\;\mathrm{CN}\left(0,\sigma_{b}^{2}\right)$ is the noise generated in the down-conversion process of the received signal [8]. Since in practice, the power of the noise generated at the antenna $\sigma_{a}^{2}$ is generally much smaller than the noise generated in the down-conversion process of the received signal $\sigma_{b}^{2}$, to simplify the derivation, the noise term $\sigma_{a}^{2}$ will be neglected in the following analysis. This kind of assumption has been wildly used in SWIPT system analysis, such as [9,10,11,12,13].

During the BC phase, the relay exploits one of the decoding methods—physical layer network coding—to decode $y_{\mathrm{ID}}$, and the decoded information is written as $x_{R}=x_{1}⊕x_{2}$. Then, the relay exhausts the harvested power $P_{R}$ to broadcast $x_{R}$. The received signal at $S_{i},i\in 1,2$ is given by:(4)$$\begin{array}{c} y_{S_{i}}=h_{i}\sqrt{P_{R}}x_{R}+n_{i} \end{array}$$where $P_{R}=\frac{2E}{T}=\eta \rho (|h_{1}{|}^{2}P_{1}+|h_{2}{|}^{2}P_{2})$, $n_{i}\;∼\;\mathrm{CN}\left(0,\sigma_{i}^{2}\right),\;i=1,2,$ is the noise generated at source $S_{i}$. For the noise components in the source and relay nodes, it is reasonable to assume $\sigma_{1}^{2}=\sigma_{2}^{2}=\sigma_{b}^{2}=\sigma^{2}$.

Finally, the source node $S_{i},\;i\in =\left\{i,2\right\}$ decodes $x_{R}$ and derives the information from $S_{j},\;j\in \left\{1,2|j\neq i\right\}$ by using its own information. For example, $S_{1}$ decodes the information from $S_{2}$ as $x_{2}=x_{R}⊕x_{1}$.

### 2.2. Performance Metric

The network outage probability is used as the performance metric. The outage probability of the network we considered is defined as the probability that the achievable instantaneous received rate pair $\left(R_{12},R_{21}\right)$ falls below an outage rate pair threshold $\left(R_{\mathrm{th}1},\;R_{\mathrm{th}2}\right)$. The achievable instantaneous received rate pair is constrained by the achievable rate regions D, which can be obtained by using the results in [14] (Refer to Equations (37)–(39) in reference [14]). The rate region is mathematically expressed as:(5)$$\begin{array}{cc} D & =\{\left(R_{12},R_{21}\right)|0\leq R_{12}\leq \min\left(R_{1R},R_{R2}\right),\\ 0 & \leq R_{21}\leq \min\left(R_{2R},R_{R1}\right),\;\;R_{12}+R_{21}\leq R_{\mathrm{MA}}\} \end{array}$$where $R_{ij},i,j\in \left[1,2,R\right]$ denotes the end-to-end transmission rate from node *i* to node *j*, $R_{\mathrm{MA}}$ denotes the MA information transfer rate region. The transmission rate $R_{ij},i,j\in \left[1,2,R\right]$ and $R_{\mathrm{MA}}$ can be calculated by $R_{ij}=\frac{1}{2}\log_{2}\left(1+\gamma_{ij}\right)$ and $R_{\mathrm{MA}}=\frac{1}{2}\log_{2}\left(1+\gamma_{\mathrm{MA}}\right)$, respectively, where $\gamma_{ij}$, $\gamma_{\mathrm{MA}}$ are the related signal-to-noise ratios (SNR).

From Equations (Equation 3) and (Equation 4), we can obtain the SNR of each transmission as the following equations:
(6a)$$\begin{array}{cc} \gamma_{1R} & =\left(1-\rho \right)\alpha k_{1} \end{array}$$
(6b)$$\begin{array}{cc} \gamma_{R2} & =|h_{2}{|}^{2}\eta \rho \left(\alpha k_{1}+\left(1-\alpha \right)k_{2}\right) \end{array}$$
(6c)$$\begin{array}{cc} \gamma_{2R} & =\left(1-\rho \right)\left(1-\alpha \right)k_{2} \end{array}$$
(6d)$$\begin{array}{cc} \gamma_{R1} & =|h_{1}{|}^{2}\eta \rho \left(\alpha k_{1}+\left(1-\alpha \right)k_{2}\right) \end{array}$$
(6e)$$\begin{array}{cc} \gamma_{\mathrm{MA}} & =\left(1-\rho \right)(\alpha k_{1}+\left(1-\alpha \right)k_{2}) \end{array}$$
where $k_{1}=\frac{|h_{1}{|}^{2}P_{t}}{\sigma^{2}}$, $k_{2}=\frac{|h_{2}{|}^{2}P_{t}}{\sigma^{2}}$, α is the power allocation ratio and $P_{1}=\alpha P_{t}$, $P_{2}=\left(1-\alpha \right)P_{t}$.

With the required target transmission rate threshold $R_{\mathrm{th}1}$ and $R_{\mathrm{th}2}$ at the receiving nodes $S_{1}$ and $S_{2}$, respectively, the outage probability can be expressed as:(7)$$\begin{array}{c} P_{\mathrm{out}}=\mathrm{Pr}\left[(R_{\mathrm{th}1},R_{\mathrm{th}2})\notin D\right]\\ =\mathrm{Pr}\left[R_{12}\;<\;R_{\mathrm{th}1}\right]\cup \mathrm{Pr}\left[R_{21}\;<\;R_{\mathrm{th}2}\right]\cup \mathrm{Pr}\left[R_{\mathrm{MA}}\;<\;R_{\mathrm{th}1}\;+\;R_{\mathrm{th}2}\right]\\ =\mathrm{Pr}\left[\min\left(R_{1R},R_{R2}\right)<R_{\mathrm{th}1}\right]\cup \mathrm{Pr}\left[\min\left(R_{2R},R_{R1}\right)<R_{\mathrm{th}2}\right]\\ \;\;\;\;\cup \mathrm{Pr}\left[R_{\mathrm{MA}}\;<\;R_{\mathrm{th}1}\;+\;R_{\mathrm{th}2}\right]\\ =\mathrm{Pr}\left[\min\left(\gamma_{1R},\gamma_{R2}\right)<\gamma_{\mathrm{th}1}\right]\cup \mathrm{Pr}\left[\min\left(\gamma_{2R},\gamma_{R1}\right)<\gamma_{\mathrm{th}2}\right]\\ \;\;\;\;\cup \mathrm{Pr}\left[\gamma_{\mathrm{MA}}<\gamma_{\mathrm{th}\Sigma }\right]\\ =\mathrm{Pr}\left[\min\left(\frac{\gamma_{1R}}{\gamma_{\mathrm{th}1}},\frac{\gamma_{R2}}{\gamma_{\mathrm{th}1}},\frac{\gamma_{2R}}{\gamma_{\mathrm{th}2}},\frac{\gamma_{R1}}{\gamma_{\mathrm{th}2}},\frac{\gamma_{\mathrm{MA}}}{\gamma_{\mathrm{th}\Sigma }}\right)<1\right] \end{array}$$where $\gamma_{\mathrm{th}i}=2^{2R_{\mathrm{th}i}}-1,\;i=1,2$, $\gamma_{\mathrm{th}\Sigma }=2^{2(R_{\mathrm{th}1}+R_{\mathrm{th}2})}-1$.

Denote $F_{X}\left(\cdot \right)$ as the cumulative distribution function (CDF) of a random variable *X*. We can rewrite Equation (Equation 7) as:(8)$$P_{\mathrm{out}}=F_{\Lambda }\left(1\right).$$where:(9)$$\begin{array}{cc} \;\Lambda \;=\; & \min\left(\frac{\gamma_{1R}}{\gamma_{\mathrm{th}1}},\frac{\gamma_{R2}}{\gamma_{\mathrm{th}1}},\frac{\gamma_{2R}}{\gamma_{\mathrm{th}2}},\frac{\gamma_{R1}}{\gamma_{\mathrm{th}2}},\frac{\gamma_{\mathrm{MA}}}{\gamma_{\mathrm{th}\Sigma }}\right)\\ \;=\; & \min\left(\frac{\left(1\;-\;\rho \right)\alpha k_{1}}{\gamma_{\mathrm{th}1}},\;\frac{|h_{2}{|}^{2}\eta \rho \left(\alpha k_{1}\;+\;\left(1-\alpha \right)k_{2}\right)}{\gamma_{\mathrm{th}1}},\right.\\ \;\;\; & \left.\frac{\left(1-\rho \right)\left(1-\alpha \right)k_{2}}{\gamma_{\mathrm{th}2}},\frac{|h_{1}{|}^{2}\eta \rho \left(\alpha k_{1}\;+\;\left(1-\alpha \right)k_{2}\right)}{\gamma_{\mathrm{th}2}},\right.\\ \;\;\; & \left.\frac{\left(1\;\;-\;\;\rho \right)(\alpha k_{1}+\left(1-\alpha \right)k_{2})}{\gamma_{\mathrm{th}\Sigma }}\;\right)\;. \end{array}$$

### 2.3. Problem Formulation

Our goal is to provide insights into the optimal PS at the relay and the optimal PA at each source node for the proposed SWIPT-TWR network to minimize the network outage probability derived in Equation (Equation 7). That is, the optimization problem is formulated as:(10)$$\begin{array}{cc} \mathrm{OP}0:\; & \left(\alpha^{o},\rho^{o}\right)=\arg\min_{\alpha ,\;\rho }\;\;P_{\mathrm{out}}\left(\alpha ,\rho \right)\\ \mathrm{subject}\mathrm{to} & \;\;0<\alpha <1\;\;\mathrm{and}\;\;0<\rho <1. \end{array}$$

With the available instantaneous channel state information (CSI), it is more desirable to formulate an equivalent joint optimization problem, which aims at maximizing the normalized SNR Λ shown in Equation (Equation 9). This transformed optimization problem is expressed as:(11)$$\begin{array}{cc} \mathrm{OP}1:\; & \left(\alpha^{o},\rho^{o}\right)=\arg\max_{\alpha ,\;\rho }\;\;\Lambda \left(\alpha ,\rho \right)\\ \mathrm{subject}\mathrm{to} & \;\;0<\alpha <1\;\;\mathrm{and}\;\;0<\rho <1. \end{array}$$

Since Λ is a complex minimization function of two variables α and ρ, it is very difficult, if not impossible, to solve for a solution directly based on the optimization formed in (Equation 11), but Λ(α,ρ) in Equation (Equation 9) is observed to be a concave function of ρ for a fixed α. This observation leads to a two-step approach to solve this joint resource allocation problem, which is developed in the next section.

## 3. A Two-Step Optimization Algorithm

To solve Equation (Equation 11) effectively, we adopt a successive approach to transform the original problem into two subproblems. First, an optimal PS ratio is obtained by fixing α. Then, the resulting PS ratio is substituted into Equation (Equation 11) to derive a closed-form solution of the optimal PA ratio through case studies. This kind of method, which transforms a complex original problem into a series of easily-solved convex problems, is a typical non-convex problem solution, which has been widely used in solving mathematical problems [15] and solving formulated optimization problems applied in communication scenarios [16].

For a fixed α, the optimization problem reduces to a one-dimensional problem related to the power splitting ratio ρ. For ∀α, the optimization problem can be expressed as:(12)$$\begin{array}{cc} \mathrm{OP}2:\; & \rho^{o}=\arg\max_{\rho }\;\;\Lambda \left(\rho \right)\\ \mathrm{subject}\mathrm{to} & 0<\rho <1. \end{array}$$where Λ(ρ) is rewritten as:(13)$$\begin{array}{c} \;\;\;\;\Lambda \left(\rho \right)=\min\left(\left(1-\rho \right)g_{1}\left(\alpha \right),\;\rho g_{2}\left(\alpha \right)\right) \end{array}$$and $g_{1}\left(\alpha \right)$, $g_{2}\left(\alpha \right)$ are expressed as:
(14a)$$\begin{array}{cc} g_{1}\left(\alpha \right) & =\min\left(\frac{\alpha k_{1}}{\gamma_{\mathrm{th}1}},\frac{\left(1-\alpha \right)k_{2}}{\gamma_{\mathrm{th}2}},\frac{\alpha k_{1}+\left(1-\alpha \right)k_{2}}{\gamma_{\mathrm{th}\Sigma }}\right) \end{array}$$
(14b)$$\begin{array}{cc} g_{2}\left(\alpha \right) & =\min\left(\frac{|h_{2}{|}^{2}}{\gamma_{\mathrm{th}1}},\frac{|h_{1}{|}^{2}}{\gamma_{\mathrm{th}2}}\right)\cdot \left[\eta \left(\alpha k_{1}+\left(1-\alpha \right)k_{2}\right)\right] \end{array}$$

It is easy to notice that Equation (Equation 13) is a convex function of ρ for a given α. Thus, the optimal ρ derived from Equation (Equation 13) is obtained when $\left(1-\rho \right)g_{1}\left(\alpha \right)=\rho g_{2}\left(\alpha \right)$, i.e.,
(15)$$\rho^{o}=\frac{g_{1}\left(\alpha \right)}{g_{1}\left(\alpha \right)+g_{2}\left(\alpha \right)}.$$

Note that $\rho^{o}$ is a function of α. Substituting Equation (Equation 15) into Λ leads to a one-dimensional function of α written as:(16)$$\Lambda \left(\alpha \right)=\frac{g_{1}\left(\alpha \right)g_{2}\left(\alpha \right)}{g_{1}\left(\alpha \right)+g_{2}\left(\alpha \right)}.$$

The optimal power allocation ratio $\alpha^{o}$ can be obtained by maximizing Equation (Equation 16), which is equivalent to minimizing its reciprocal transformation $\bar{\Lambda }\left(\alpha \right)=\left(\frac{1}{g_{1}\left(\alpha \right)}+\frac{1}{g_{2}\left(\alpha \right)}\right)$. The transformed problem is written as follows.
(17)$$\begin{array}{cc} \mathrm{OP}3:\; & \alpha^{o}=\arg\min\bar{\Lambda }\left(\alpha \right)\\ & \;\;\;=\arg\min\left(\frac{1}{g_{1}\left(\alpha \right)}+\frac{1}{g_{2}\left(\alpha \right)}\right)\\ \mathrm{subject}\mathrm{to} & 0<\alpha <1 \end{array}$$

Equation (Equation 17) shows that the optimal $\alpha^{o}$ is determined by $g_{1}\left(\alpha \right)$ and $g_{2}\left(\alpha \right)$. Define the three components of $g_{1}\left(\alpha \right)$ in Equation ([Disp-formula FD14a-sensors-18-03946]) as $f_{1}=\alpha k_{1}/\gamma_{\mathrm{th}1}$, $f_{2}=\left(1-\alpha \right)k_{2}/\gamma_{\mathrm{th}2}$ and $f_{3}=\left(\alpha k_{1}+\left(1-\alpha \right)k_{2}\right)/\gamma_{\mathrm{th}\Sigma }$. Let $\alpha_{12}$, $\alpha_{13}$ and $\alpha_{23}$ be, respectively, the points of intersection of $f_{1}$ and $f_{2}$, $f_{1}$ and $f_{3}$ and $f_{2}$ and $f_{3}$. The analytical expression of $\bar{\Lambda }\left(\alpha \right)$ can be classified as two cases, which is shown in the following Theorem 1.

**Theorem** **1.**
*$\bar{\Lambda }\left(\alpha \right)$ is a continuous piecewise function and can be divided into two cases dependent on the size of $\alpha_{13}$ and $\alpha_{23}$. The two cases are:*

*Case 1: When $\alpha_{13}\geq \alpha_{23}$:*
(18)$$\bar{\Lambda }\left(\alpha \right)=\left\{\;\;\;\begin{array}{c} \frac{1}{f_{1}}+\frac{1}{v}\cdot {\left(\alpha k_{1}+\left(1-\alpha \right)k_{2}\right)}^{-1},\;\;0<\alpha \leq \alpha_{12}\\ \frac{1}{f_{2}}+\frac{1}{v}\cdot {\left(\alpha k_{1}+\left(1-\alpha \right)k_{2}\right)}^{-1},\;\;\alpha_{12}\leq \alpha <1 \end{array}\right.$$

*Case 2: When $\alpha_{13}<\alpha_{23}$:*
(19)$$\bar{\Lambda }\left(\alpha \right)=\left\{\;\;\;\begin{array}{c} \frac{1}{f_{1}}+\frac{1}{v}\cdot {\left(\alpha k_{1}+\left(1-\alpha \right)k_{2}\right)}^{-1},\;\;0<\alpha \leq \alpha_{13}\\ \frac{1}{f_{3}}+\frac{1}{v}\cdot {\left(\alpha k_{1}+\left(1-\alpha \right)k_{2}\right)}^{-1},\;\;\alpha_{13}\leq \alpha \leq \alpha_{23}\\ \frac{1}{f_{2}}+\frac{1}{v}\cdot {\left(\alpha k_{1}+\left(1-\alpha \right)k_{2}\right)}^{-1},\;\;\alpha_{23}\leq \alpha <1 \end{array}\right.$$
*where $v=\eta \cdot \min\left(\frac{|h_{2}{|}^{2}}{\gamma_{\mathrm{th}1}},\frac{|h_{1}{|}^{2}}{\gamma_{\mathrm{th}2}}\right)$,*
(20)$$\begin{array}{cc} \alpha_{12} & =\frac{k_{2}\gamma_{\mathrm{th}1}}{k_{1}\gamma_{\mathrm{th}2}+k_{2}\gamma_{\mathrm{th}1}}\\ \alpha_{13} & =\frac{k_{2}\gamma_{\mathrm{th}1}}{k_{1}\left(\gamma_{\mathrm{th}\Sigma }-\gamma_{\mathrm{th}1}\right)+k_{2}\gamma_{\mathrm{th}1}}\\ \alpha_{23} & =\frac{k_{2}\left(\gamma_{\mathrm{th}\Sigma }-\gamma_{\mathrm{th}2}\right)}{k_{2}\left(\gamma_{\mathrm{th}\Sigma }-\gamma_{\mathrm{th}2}\right)+k_{1}\gamma_{\mathrm{th}2}} \end{array}$$


**Proof.** The proof is given in Appendix A. ☐

It can be concluded that the monotony of $f_{3}=\left(\alpha k_{1}+\left(1-\alpha \right)k_{2}\right)/\gamma_{\mathrm{th}\Sigma }$, $g_{2}\left(\alpha \right)=v\cdot \left(\alpha k_{1}+\left(1-\alpha \right)k_{2}\right)$ has a relationship with the size of $k_{1}$ and $k_{2}$. When $k_{1}=k_{2}$, $f_{3}$ and $g_{2}\left(\alpha \right)$ are constant; when $k_{1}>k_{2}$, $f_{3}$ and $g_{2}\left(\alpha \right)$ are monotonically increasing functions of α; and when $k_{1}<k_{2}$, $f_{3}$ and $g_{2}\left(\alpha \right)$ are monotonically decreasing functions of α. Thus, the monotony of $\bar{\Lambda }\left(\alpha \right)$ for each case in Theorem 1 has three subcases. With the above analysis, we analyze **OP3** with case studies.

### 3.1. Case 1: $\alpha_{13}\geq \alpha_{23}$

In this case, $\bar{\Lambda }\left(\alpha \right)$ is a continuous two-segment piecewise function generated from Equation (Equation 18), With the size of $k_{1}$ and $k_{2}$, $\bar{\Lambda }\left(\alpha \right)$ and **OP3** have three subcases, which are analyzed as follows.

#### 3.1.1. Subcase 1: $k_{1}=k_{2}$

In this subcase, $g_{2}\left(\alpha \right)$ is a constant for all values of α; thus, the variation of $\bar{\Lambda }\left(\alpha \right)$ is determined by $f_{1}$ or $f_{2}$. From Equation (Equation 18), it is easy to get that, in the range of $\alpha \in (0,\alpha_{12}]$, $\bar{\Lambda }\left(\alpha \right)={f_{1}}^{-1}+v^{-1}\cdot {\left(\alpha k_{1}+\left(1-\alpha \right)k_{2}\right)}^{-1}$ is a decreasing function of α; and in the range of $\alpha \in (\alpha_{12},1)$, $\bar{\Lambda }\left(\alpha \right)={f_{2}}^{-1}+v^{-1}\cdot {\left(\alpha k_{1}+\left(1-\alpha \right)k_{2}\right)}^{-1}$ is an increasing function of α. Thus, in this subcase, $\bar{\Lambda }\left(\alpha \right)$ is non-negative convex, which gets its minimum at $\alpha =\alpha_{12}$. The solution of OP3occurs at $\alpha =\alpha_{12}$.

#### 3.1.2. Subcase 2: $k_{1}>k_{2}$

In this subcase, $g_{2}\left(\alpha \right)$ is an increasing function, and $f_{3}$ does not affect $\bar{\Lambda }\left(\alpha \right)$. Through analysis, Lemma 1 can be obtained as follows.

**Lemma** **1.**
*The optimal power allocation of Subcase 2 is unique and lies in the range of $\alpha \in [\alpha_{12},1]$, which can be calculated by $\alpha^{o}=\alpha_{\mathrm{case}1}^{*}=\max\left(\alpha^{*},\alpha_{12}\right)$. $\alpha^{*}$ is the stationary point of the second segment of $\bar{\Lambda }\left(\alpha \right)$, which is obtained by solving ${\left(\bar{\Lambda }\left(\alpha \right)\right)}^{\prime }={\left({f_{2}}^{-1}+v^{-1}\cdot {\left(\alpha k_{1}+\left(1-\alpha \right)k_{2}\right)}^{-1}\right)}^{\prime }=0$, and the solved $\alpha^{*}$ is:*
(21)$$\begin{array}{c} \alpha^{*}=\frac{\sqrt{k_{2}\left(k_{1}-k_{2}\right)}-k_{2}\sqrt{v\gamma_{\mathrm{th}2}}}{\sqrt{k_{2}\left(k_{1}-k_{2}\right)}+\left(k_{1}-k_{2}\right)\sqrt{v\gamma_{\mathrm{th}2}}} \end{array}$$


**Proof.** With the former analysis that $g_{1}\left(\alpha \right)$ is a non-negative convex function with its maxima at $\alpha =\alpha_{12}$ and $g_{2}\left(\alpha \right)$ is an increasing function in this subcase, it is easy to see that $\bar{\Lambda }\left(\alpha =\alpha_{12}\right)$ is the minima in the range of $\alpha \in [0,\alpha_{12}]$. Given $\bar{\Lambda }\left(\alpha_{12}\right)$ as the initial value of $\bar{\Lambda }$ in the range of $\alpha \in [\alpha_{12},1]$, the variation of $\bar{\Lambda }\left(\alpha \right)$ in the range $\alpha \in [\alpha_{12},1]$ is analyzed as follows. There is the conclusion that $g_{1}\left(\alpha \right)=f_{1}$ decreases as α increases and $g_{2}\left(\alpha \right)$ increases as α increases in the range of $\alpha \in [\alpha_{12},1]$. This characteristic leads to the conclusion that the optimal value of $\bar{\Lambda }$ occurs in the range of $\left[\alpha_{12},1\right]$. In this range, $\bar{\Lambda }\left(\alpha \right)=\frac{1}{f_{2}}+\frac{1}{g_{2}}$. The second-order derivative of $\bar{\Lambda }\left(\alpha \right)$ given in Equation ([Disp-formula FD22a-sensors-18-03946]) is greater than zero; thus, the optimal α can be obtained by solving $\frac{\partial \bar{\Lambda }\left(\alpha \right)}{\partial \alpha }={\left[\frac{1}{f_{2}}+\frac{1}{v}\cdot {\left(\alpha k_{1}+\left(1-\alpha \right)k_{2}\right)}^{-1}\right]}_{\alpha }^{{}^{\prime }}=0$. The first-order derivative $\frac{\partial \bar{\Lambda }\left(\alpha \right)}{\partial \alpha }$ is given in Equation ([Disp-formula FD22b-sensors-18-03946]).
(22a)$$\begin{array}{cc} \frac{\partial^{2}\bar{\Lambda }\left(\alpha \right)}{{\partial \alpha }^{2}} & =\frac{2\gamma_{\mathrm{th}2}}{k_{2}}\frac{1}{{\left(1-\alpha \right)}^{3}}+\frac{1}{v}\frac{{\left(k_{1}-k_{2}\right)}^{2}}{{\left(\alpha k_{1}+\left(1-\alpha \right)k_{2}\right)}^{4}} \end{array}$$
(22b)$$\begin{array}{cc} \frac{\partial \bar{\Lambda }\left(\alpha \right)}{\partial \alpha } & =\frac{\gamma_{\mathrm{th}2}}{k_{2}}\frac{1}{{\left(1-\alpha \right)}^{2}}-\frac{1}{v}\frac{k_{1}-k_{2}}{{\left(\alpha k_{1}+\left(1-\alpha \right)k_{2}\right)}^{2}}. \end{array}$$ ☐

By solving Equation ([Disp-formula FD22b-sensors-18-03946]), one obtains the minimum of ${\left[\frac{1}{f_{2}}+\frac{1}{v}\cdot {\left(\alpha k_{1}+\left(1-\alpha \right)k_{2}\right)}^{-1}\right]}_{\alpha }^{{}^{\prime }}$, which is shown in Equation (Equation 21). Compared with the boundary value in the range of $\alpha \in [\alpha_{12},1]$, we get the conclusion that if $\alpha^{*}<\alpha_{12}$, the optimal power allocation ratio is $\alpha^{o}=\alpha_{12}$; else, if $\alpha^{*}\geq \alpha_{12}$, the optimal power allocation ratio is $\alpha^{o}=\alpha^{*}$. Rephrasing the above analysis results in Lemma 1.

#### 3.1.3. Subcase 3: $k_{1}<k_{2}$

In this subcase, $g_{2}\left(\alpha \right)$ is a decreasing function and $f_{3}$ does not affect $\bar{\Lambda }\left(\alpha \right)$. Through analysis, Lemma 2 can be derived as follows.

**Lemma** **2.**
*The optimal power allocation of Subcase 3 is unique and lies in the range of $\alpha \in [0,\alpha_{12}]$, which can be calculated by $\alpha^{o}=\alpha_{\mathrm{case}1}^{+}=\min\left(\alpha^{+},\alpha_{12}\right)$. $\alpha^{+}$ is the stationary point of the first segment of $\bar{\Lambda }\left(\alpha \right)$, which is obtained by solving ${\left(\bar{\Lambda }\left(\alpha \right)\right)}^{\prime }={\left({f_{1}}^{-1}+v^{-1}\cdot {\left(\alpha k_{1}+\left(1-\alpha \right)k_{2}\right)}^{-1}\right)}^{\prime }=0$, and the solved $\alpha^{+}$ is:*
(23)$$\begin{array}{c} \alpha^{+}=\frac{k_{2}\sqrt{v\cdot \gamma_{\mathrm{th}1}}}{\sqrt{k_{2}-k_{1}}\left(\sqrt{k_{1}}+\sqrt{v\cdot \gamma_{\mathrm{th}1}\left(k_{2}-k_{1}\right)}\right)} \end{array}$$


**Proof.** The convexity of this case can be verified using a similar analysis as Lemma 1. Given $\alpha_{12}$ as an initial value, it is easy to see that $g_{1}\left(\alpha \right)$ decreases as α decreases from this initial value; however, $g_{2}\left(\alpha \right)$ increases as α decreases from this initial value. This characteristic shows that the optimal value of $\bar{\Lambda }\left(\alpha \right)$ occurs in the range of $\left[0,\alpha_{12}\right]$, where $g_{1}\left(\alpha \right)=f_{1}\left(\alpha \right)$. Thus, $\bar{\Lambda }\left(\alpha \right)=\frac{1}{f_{1}}+v^{-1}\cdot {\left(\alpha k_{1}+\left(1-\alpha \right)k_{2}\right)}^{-1}$. Since the second-order derivative of $\bar{\Lambda }\left(\alpha \right)$ given in Equation ([Disp-formula FD24a-sensors-18-03946]) is greater than zero, the optimal α is obtained by letting the first derivative $\frac{\partial \bar{\Lambda }\left(\alpha \right)}{\partial \alpha }={\left[\frac{1}{f_{1}}+v^{-1}\cdot {\left(\alpha k_{1}+\left(1-\alpha \right)k_{2}\right)}^{-1}\right]}_{\alpha }^{{}^{\prime }}$ expressed in ([Disp-formula FD24b-sensors-18-03946]) equal zero.
(24a)$$\begin{array}{cc} \frac{\partial^{2}\bar{\Lambda }\left(\alpha \right)}{{\partial \alpha }^{2}} & =\frac{2\gamma_{\mathrm{th}1}}{k_{1}}\frac{1}{\alpha^{3}}+\frac{1}{v}\frac{{\left(k_{1}-k_{2}\right)}^{2}}{{\left(\alpha k_{1}+\left(1-\alpha \right)k_{2}\right)}^{4}} \end{array}$$
(24b)$$\begin{array}{cc} \frac{\partial \bar{\Lambda }\left(\alpha \right)}{\partial \alpha } & =-\frac{\gamma_{\mathrm{th}1}}{k_{1}}\frac{1}{\alpha^{2}}-\frac{1}{v}\frac{k_{1}-k_{2}}{{\left(\alpha k_{1}+\left(1-\alpha \right)k_{2}\right)}^{2}} \end{array}$$ ☐

By solving Equation ([Disp-formula FD24b-sensors-18-03946]), one obtains the minimum of ${\left[\frac{1}{f_{1}}+\frac{1}{v}\cdot {\left(\alpha k_{1}+\left(1-\alpha \right)k_{2}\right)}^{-1}\right]}_{\alpha }^{{}^{\prime }}$, which is shown in Equation (Equation 23). Compared with the boundary value in the range of $\alpha \in [0,\alpha_{12}]$, we get the conclusion that if $\alpha^{+}<\alpha_{12}$, the optimal power allocation ratio is $\alpha^{o}=\alpha^{+}$; else, if $\alpha^{+}\geq \alpha_{12}$, the optimal power allocation ratio is $\alpha^{o}=\alpha_{12}$. Rephrasing the above analysis results in Lemma 2.

### 3.2. Case 2: $\alpha_{13}<\alpha_{23}$

In this case, Λ(α) is generated from Equation (Equation 19), which is a three-segment continuous function. Since $f_{3}$ and $g_{2}\left(\alpha \right)$ have a relationship with the size of the channel gains $k_{1}$ and $k_{2}$, the same as for Case 1, Case 2 has three subcases, as well.

#### 3.2.1. Subcase 1: $k_{1}=k_{2}$

In this subcase, $f_{3}$ and $g_{2}\left(\alpha \right)$ are constants for all values of α. From Equation (Equation 19), it is easy to get that, the first segment of $\bar{\Lambda }\left(\alpha \right)$ is a decreasing function, the third segment of $\bar{\Lambda }\left(\alpha \right)$ is an increasing function and the second segment of $\bar{\Lambda }\left(\alpha \right)$ is a constant. Thus, $\bar{\Lambda }$ will get its minimum at $\forall \alpha \in [\alpha_{13},\alpha_{23}]$.

#### 3.2.2. Subcase 2: $k_{1}>k_{2}$

In this subcase, $f_{3}$ and $g_{2}\left(\alpha \right)$ are monotonically increasing functions. Through analysis, Lemma 3 can be obtained as follows.

**Lemma** **3.**
*The optimal power allocation for this subcase is unique and lies in the range of $\alpha \in [\alpha_{23},1)$, which can be calculated by $\alpha^{o}=\alpha_{\mathrm{case}2}^{*}=\max\left(\alpha^{*},\alpha_{23}\right)$. $\alpha^{*}$ is the stationary point of the third segment of $\bar{\Lambda }\left(\alpha \right)$, which is equivalent to that obtained in Equation (Equation 21).*


**Proof.** Since $f_{1}$ and $f_{2}$ are respectively an increasing function and a decreasing function with the increase of α, since $f_{3}$ and $g_{2}\left(\alpha \right)$ are monotonic functions in this subcase, it is easy to obtain that the first and second segments of $\bar{\Lambda }\left(\alpha \right)$ are decreasing functions. Due to the continuous feature, $\bar{\Lambda }\left(\alpha =\alpha_{23}\right)$ is the minima in the range of $\alpha \in [0,\alpha_{23}]$. Given $\bar{\Lambda }\left(\alpha_{23}\right)$ as the initial value of $\bar{\Lambda }$ in the range of $\alpha \in [\alpha_{23},1]$, the variation of $\bar{\Lambda }\left(\alpha \right)$ in the range of $\alpha \in [\alpha_{23},1]$ is analyzed as follows. In the range, $\bar{\Lambda }\left(\alpha \right)=\frac{1}{f_{2}}+\frac{1}{g_{2}}$. The second-order derivation and first-order derivation of $\bar{\Lambda }\left(\alpha \right)$ can be obtained as shown in Equations ([Disp-formula FD22a-sensors-18-03946]) and ([Disp-formula FD22b-sensors-18-03946]). $\alpha^{*}$ is the minima of ${\left[\frac{1}{f_{2}}+\frac{1}{v}\cdot {\left(\alpha k_{1}+\left(1-\alpha \right)k_{2}\right)}^{-1}\right]}_{\alpha }^{{}^{\prime }}$, which is shown in Equation (Equation 21). Compared with the boundary value in the range of $\alpha \in [\alpha_{23},1]$, we get the conclusion that if $\alpha^{*}<\alpha_{23}$, the optimal power allocation value is $\alpha^{o}=\alpha_{23}$; else, if $\alpha^{*}\geq \alpha_{23}$, the optimal power allocation value is $\alpha^{o}=\alpha^{*}$. Rephrasing the above analysis results in Lemma 3. ☐

#### 3.2.3. Subcase 3: $k_{1}<k_{2}$

In this subcase, $f_{3}$ and $g_{2}\left(\alpha \right)$ are monotonically decreasing functions. Through analysis, Lemma 4 can be obtained as follows.

**Lemma** **4.**
*The optimal power allocation for this subcase is unique and lies in the range of $\alpha \in (0,\alpha_{13}]$, which can be calculated by $\alpha^{o}=\alpha_{\mathrm{case}2}^{+}=\min\left(\alpha^{+},\alpha_{13}\right)$. $\alpha^{+}$ is the stationary point of the first segment of $\bar{\Lambda }\left(\alpha \right)$, which is equivalent to that obtained in Equation (Equation 23).*


**Proof.** With monotonically decreasing functions of $f_{3}$ and $g_{2}\left(\alpha \right)$, the second and third segments of $\bar{\Lambda }\left(\alpha \right)$ are increasing functions. Due to the continuous feature, $\bar{\Lambda }\left(\alpha =\alpha_{13}\right)$ is the minima in the range of $\alpha \in [\alpha_{13},1]$; whereas in the range of $\alpha \in [0,\alpha_{13}]$, $\bar{\Lambda }\left(\frac{1}{f_{1}}\right)+\frac{1}{v}\cdot \left(\alpha k_{1}+\left(1-\alpha \right)k_{2}\right)$. The second-order derivation and first-order derivation of $\bar{\Lambda }\left(\alpha \right)$ can be obtained as shown in Equations ([Disp-formula FD24a-sensors-18-03946]) and ([Disp-formula FD24b-sensors-18-03946]). $\alpha^{+}$ is the minima of ${\left[\frac{1}{f_{1}}+\frac{1}{v}\cdot {\left(\alpha k_{1}+\left(1-\alpha \right)k_{2}\right)}^{-1}\right]}_{\alpha }^{{}^{\prime }}$, which is shown in Equation (Equation 23). Compared with the boundary value in the range of $\alpha \in [0,\alpha_{13}]$, we get the conclusion that if $\alpha^{+}<\alpha_{13}$, the optimal power allocation value is $\alpha^{o}=\alpha^{+}$; else, if $\alpha^{+}\geq \alpha_{13}$, the optimal power allocation value is $\alpha^{o}=\alpha_{13}$. Rephrasing the above analysis results in Lemma 4. ☐

## 4. The Closed-Form of PA and PS

From the above analysis, the closed-form of optimal power allocation ratio $\alpha^{o}$ is calculated and presented below.

When $\alpha_{12}\geq \alpha_{23}$:(25)$$\alpha^{o}=\left\{\;\;\;\begin{array}{c} \alpha_{12},\;\mathrm{if}\;k_{1}=k_{2}\\ \alpha_{case1}^{*},\;\;\mathrm{if}\;k_{1}>k_{2}\\ \alpha_{case1}^{+},\;\;\mathrm{if}\;k_{1}<k_{2} \end{array}\right.$$

When $\alpha_{12}<\alpha_{23}$:(26)$$\alpha^{o}=\left\{\;\;\;\begin{array}{c} \forall \alpha \in \left[\alpha_{13},\alpha_{23}\right],\;\mathrm{if}\;k_{1}=k_{2}\\ \alpha_{case2}^{*},\;\;\mathrm{if}\;k_{1}>k_{2}\\ \alpha_{case2}^{+},\;\;\mathrm{if}\;k_{1}<k_{2} \end{array}\right.$$

Substituting $\alpha^{o}$ into Equation (Equation 15), the optimal power splitting ratio is obtained below.
(27)$$\begin{array}{c} \rho^{o}=\frac{g_{1}\left(\alpha^{o}\right)}{g_{1}\left(\alpha^{o}\right)+g_{2}\left(\alpha^{o}\right)} \end{array}$$

Algorithm 1 summarizes the optimal power allocation ratio and power splitting ratio design for a given set of η, $|h_{1}{|}^{2}$, $|h_{2}{|}^{2}$, $R_{\mathrm{th}1}$, $R_{\mathrm{th}2}$ and $P_{t}$.

**Algorithm 1** Optimal joint resource allocation for $\alpha^{o}$, $\rho^{o}$.
1:Given η, $|h_{1}{|}^{2}$, $|h_{2}{|}^{2}$, $R_{\mathrm{th}1}$, $R_{\mathrm{th}2}$ and $P_{t}$   2:Compute $k_{1}=\frac{|h_{1}{|}^{2}P_{t}}{\sigma^{2}}$ and $k_{2}=\frac{|h_{2}{|}^{2}P_{t}}{\sigma^{2}}$   3:Compute $\alpha_{13}$, $\alpha_{23}$.   4:
**if**
$\alpha_{13}\geq \alpha_{23}$
**then**
5: **if**
$k_{1}=k_{2}$
**then**6:  $\alpha^{o}=\alpha_{12}$   7: **else if**
$k_{1}>k_{2}$
**then**8:  Compute $\alpha^{*}$, $\alpha_{12}$   9:  $\alpha^{o}=\alpha_{case1}^{*}=\max\left(\alpha^{*},\alpha_{12}\right)$   10: **else if**
$k_{1}<k_{2}$
**then**11:  Compute $\alpha^{+}$, $\alpha_{12}$   12:  $\alpha^{o}=\alpha_{case1}^{+}=\min\left(\alpha^{+},\alpha_{12}\right)$   13: **end if**  14:
**else if**
$\alpha_{13}<\alpha_{23}$
**then**
15: **if**
$k_{1}=k_{2}$
**then**16:  $\alpha^{o}=\forall \alpha_{o}\in \left[\alpha_{13},\alpha_{23}\right]$   17: **else if**
$k_{1}>k_{2}$
**then**18:  Compute $\alpha^{*}$   19:  $\alpha^{o}=\alpha_{case2}^{*}=\max\left(\alpha^{*},\alpha_{23}\right)$   20: **else if**
$k_{1}<k_{2}$
**then**21:  Compute $\alpha^{+}$   22:  $\alpha^{o}=\alpha_{case2}^{+}=\min\left(\alpha^{+},\alpha_{13}\right)$   23: **end if**  24:
**end if**
25:Compute $\rho^{o}=\frac{g_{1}\left(\alpha^{o}\right)}{g_{1}\left(\alpha^{o}\right)+g_{2}\left(\alpha^{o}\right)}$   


## 5. Numerical Results

In this section, some numerical and simulation results are presented to verify the proposed algorithm and to assess the influence of various system parameters. To better display the superiority of the proposed resource allocation scheme, three benchmark schemes: optimal α equal ρ, optimal ρ equal α and equal α and ρ are presented as a comparison. The parameters are set as follows: the energy conversion efficiency is set to be η=0.8; the distance between the two sources is d=10; the noise power is set to be $\sigma^{2}=-90$ dBm; the channel gains are set to be $|h_{i}\left|=\right|g_{i}|\cdot {\left(1+d_{i}\right)}^{-m}$, where $g_{i}∼\mathrm{CN}\left(0,\lambda_{i}\right)$ represents the Rayleigh fading coefficient; $d_{i}$ is the distance between $S_{i}$ and R; *m* is the channel path loss exponent (m=2); and $\lambda_{1}=\lambda_{2}=1$. All results are generated from $10^{4}$ channel realizations.

Figure 2 depicts the effects of total power constraint on the system outage probability with rate threshold $\left(R_{\mathrm{th}1},R_{\mathrm{th}1}\right)=\left(1,1\right)$ bit/s/Hz. Since Algorithm 1 shows that the asymmetry of the channel affects the optimal parameters, and thus further affects the outage performance, this simulation chose two channel conditions to assess the system performance: symmetric channel condition $d_{1}=d_{2}=5$ m (the left figure); asymmetric channel condition $\left(d_{1},d_{2}\right)=\left(3,7\right)$ m (the right figure). Note that $\left(d_{1},d_{2}\right)=\left(3,7\right)$ is just a case of the asymmetric channel condition, which we chose to reflect the system performance; relay deployments of $\left(d_{1},d_{2}\right)=\left\{\left(1,9\right),\left(2,8\right),\left(4,6\right),\cdots \right\}$ are asymmetric channel conditions, as well. From this figure, we can see that the outage probability decreases with the increase of total power constraint $P_{t}$. $P_{\mathrm{out}}$ of the proposed jointly optimal resource allocation obtained by using the proposed Algorithm (A1) matches precisely with that obtained by using numerical search (NS) over the range of α∈[0,1] (step size of 0.005). The $P_{\mathrm{out}}$ of three benchmark schemes are presented, as well, which verified that the proposed jointly optimal resource allocation scheme outperforms the three benchmark schemes. The advantage of the proposed schemes is more obvious when the channel is asymmetric.

Figure 3 shows the system outage probabilities $P_{\mathrm{out}}$ versus the relay deployment. For relay deployment, the distance between source $S_{1}$ and relay *R* ($d_{1}$) is used as the *x*-axis. Simulation parameters are set as: $\gamma_{t}=15$ dBm (the left figure) or $\gamma_{t}=20$ dBm (the right figure), $\left(R_{\mathrm{th}1},R_{\mathrm{th}1}\right)=\left(1,1\right)$ bit/s/Hz. It is observed that increasing the distance between source $S_{1}$ and relay (i.e., $d_{1}$) leads to a concave characteristic of the outage probability. This behavior shows that the outage performance is much better when the relay is close to any of the source nodes than when the relay is in the middle of the two sources. It can be also noted that compared with the other three benchmark schemes, the proposed scheme has the best outage performance when the channels are asymmetric. When the channel is symmetric, the proposed scheme has the same outage performance as the scheme optimal ρ equal α. This is because when the channels are symmetric, the optimal α is equal to α with $\left(R_{\mathrm{th}1},R_{\mathrm{th}1}\right)=\left(1,1\right)\;$bit/s/Hz. Thus, to better enhance the system performance, the relay should be deployed close to each of the source nodes.

Figure 4 shows the outage performance $P_{\mathrm{out}}$ versus the rate threshold $R_{\mathrm{th}1}$ with setting $R_{\mathrm{th}2}\;=\;1$ bit/s/Hz. The simulation evaluates two situations: (a) the target transmit power is $P_{t}=15$ dBm (the left figure); (b) the target transmit power is $P_{t}=20$ dBm (the right figure). The relay is deployed at $d_{1}=3$ m and $d_{2}=7$ m. It is observed that as $R_{\mathrm{th}1}$bit/s/Hz increases, the outage probability becomes worse. This is because with the increase of the threshold rate, the achievable rate region is more prone to less than the rate threshold. Thus, the outage probability increases based on Equation (Equation 7). This figure also shows that the jointly optimal resource allocation scheme outperforms the other three benchmark schemes with any of the threshold settings.

## 6. Conclusions

We have derived a joint optimal resource allocation design for DF-TWR networks with SWIPT to minimize its system outage probability. The optimization of such a network is a very complex problem. To make it easy to tackle, a two-step method is proposed. With the two-step method, the optimal PS ratio for a given PA ratio is derived first, from which it is found that it is a function of the PA ratio. Then, the obtained PS ratio is substituted back into the main optimization problem to determine the closed-form of the optimal PA ratio. Simulation results matched the analytical results well and confirmed that the optimized system achieved a lower outage probability than existing schemes, especially in asymmetric channel conditions.

## Figures and Tables

**Figure 1 sensors-18-03946-f001:**
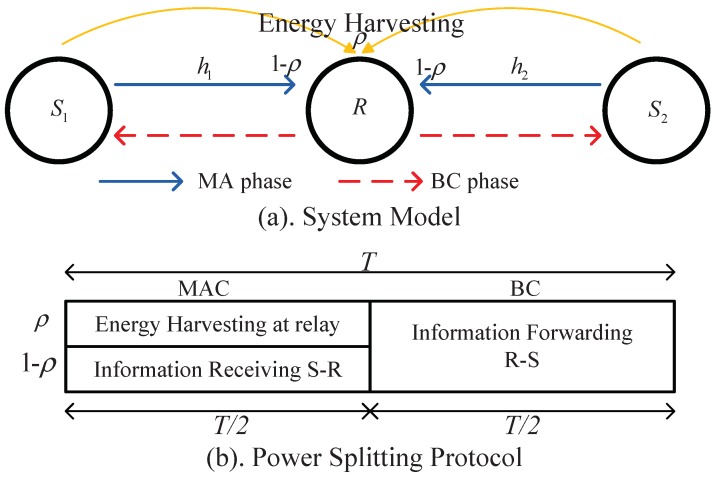
(**a**) System model (**b**) energy-harvesting protocol.

**Figure 2 sensors-18-03946-f002:**
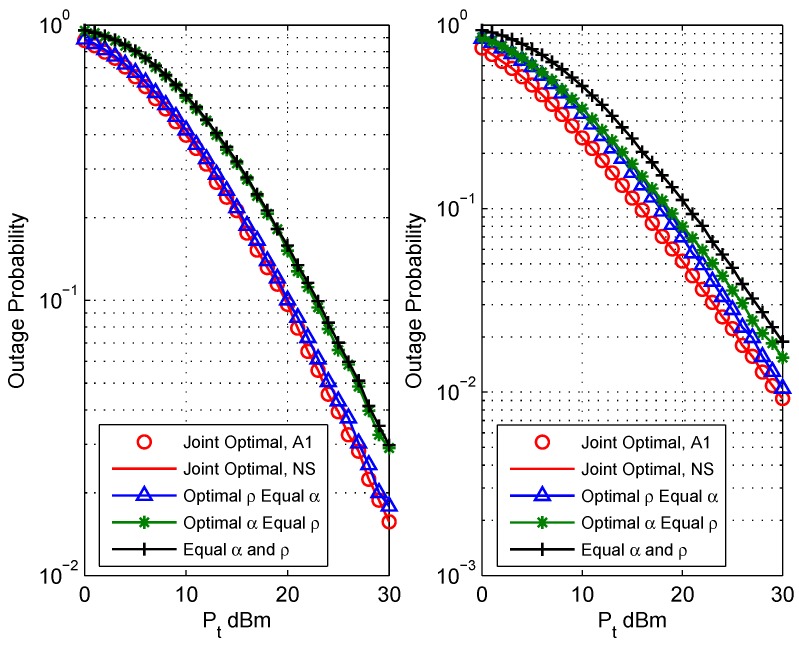
Comparison of outage performance vs. the total power constraint of different resource allocation schemes: (**a**) $\left(d_{1},d_{2}\right)=\left(5,5\right)$ (the left figure); (**b**) $\left(d_{1},d_{2}\right)=\left(3,7\right)$ (the right figure).

**Figure 3 sensors-18-03946-f003:**
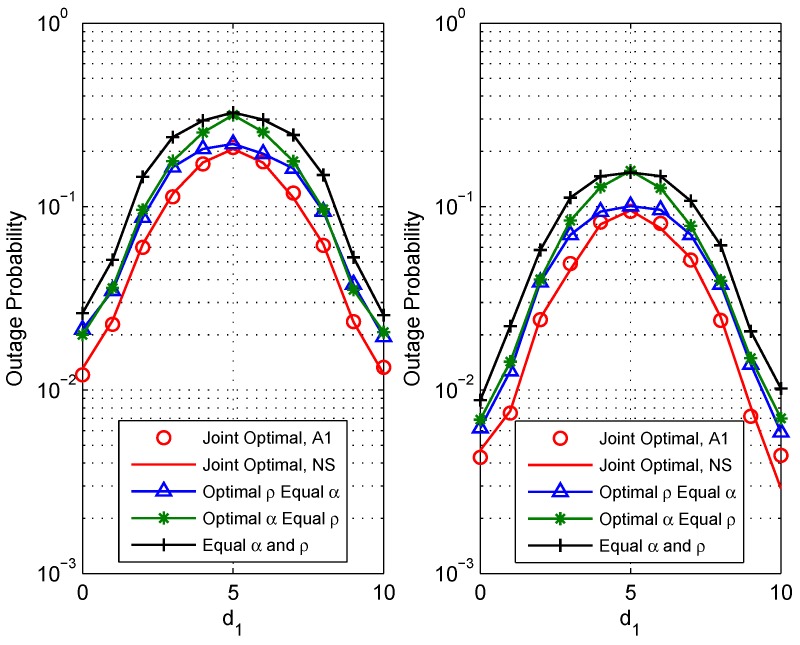
Comparison of the outage performance vs. the distance between source $S_{1}$ and relay *R* ($d_{1}$) of different schemes: (**a**) $P_{t}=15$ dBm (the left figure); (**b**)$P_{t}=20$ dBm (the right figure).

**Figure 4 sensors-18-03946-f004:**
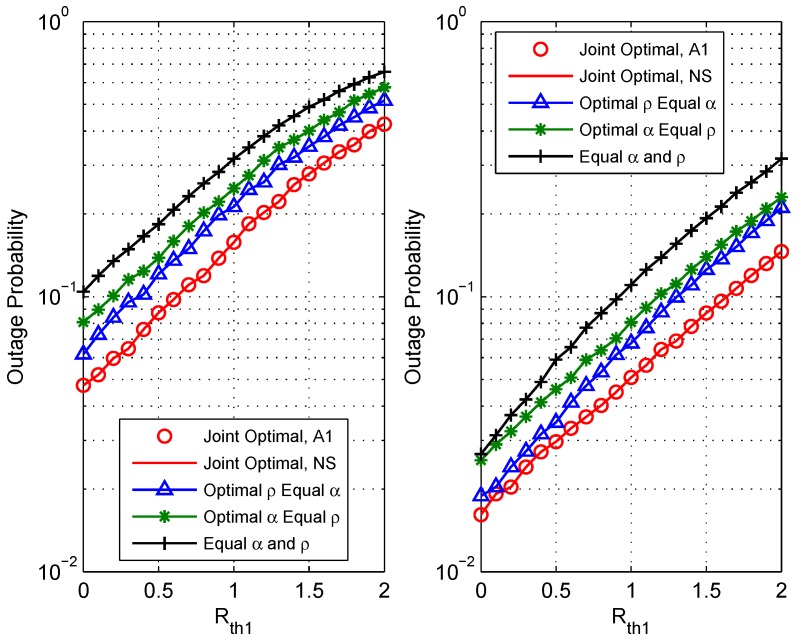
Comparison of outage performance vs. the rate threshold $R_{\mathrm{th}1}$ of different schemes: (**a**) $P_{t}\;=\;15$ dBm (the left figure); (**b**) $P_{t}=20$ dBm (the right figure).

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
