# Peer review of "Outage-Based Resource Allocation for DF Two-Way Relay Networks with Energy Harvesting"

_sensors, 2018, doi:10.3390/s18113946_

Reviewer 1 Report

The paper deals with the problem of joint resource allocation for power constrained two-way relay networks. The latter should be solved by a  joint optimization problem in which the power allocation and the power splitting are optimized. Since this joint optimization is difficult, the idea is to use an alternating minimization scheme constituted by two sub-problems having the closed formula solution.

The problem proposed in the paper is interesting, however there are some unclear points, hence the authors should clarify them. Below my concerns:

1) Alternating minimization schemes with closed form solutions for solving a joint optimization problem are not new. For instance, an alternating minimization scheme has been recently proposed for solving a factor analysis problem, see  

  [An alternating minimization algorithm for Factor

Analysis, https://arxiv.org/pdf/1806.04433.pdf]. The authors should mention that.

2) Does the proposed algorithm converge theoretically? The authors should comment on that.

For instance, in [An alternating minimization algorithm for Factor

Analysis, https://arxiv.org/pdf/1806.04433.pdf]

the convergence properties of the alternating algorithm have been analyzed.

3) It is not clear why it is possible to neglect the harvested energy from the noise \sigma^2_a

Author Response

Dear reviewer, thank you so much for the efficient review process and all the useful comments. Below are the manuscript’s revision descriptions.

1) We add the statement of reason why we choose this kind of solving method and cited corresponding references (include the one you mentioned) in the first paragraph of Sec.III.

2) The correctness of the proposed algorithm has been verified by comparing it with numerical search (NS) results which is presented in simulation part. Since the proposed algorithm obtains the closed-form analytical solution just need few comparison steps (compare \alpha_13 and \alpha_23, and compare k_1 and k_2), it does not need to analyze its convergence. Whereas the algorithm given in the reference [https://arxiv.org/pdf/1806.04433.pdf] should use interactive algorithm, so it needs to analyze its convergence.

3) We add the explanation why we neglect the noise \sigma^2_a below Eq.(3).

Reviewer 2 Report

All of figures Fig. x or Figure x. can be identified.

From line 210 The two lines obtained by both methods match precisely. This  simulation assesses two relay deployment scenarios: (a) symmetric channel condition d1 = d2 = 5m; (b) asymmetric channel condition (d1, d2) = (3, 7)m. How about (b) asymmetric channel condition (d1, d2) = (1, 9)m, (d1, d2) = (2, 8)m or (d1, d2) = (4, 6)m…

Why to select asymmetric channel condition (d1, d2) = (3, 7)m for the comparison with (a) symmetric channel condition d1 = d2 = 5m ?

How to apply the paper algorithm or methodology simulations for the practical information and communication fields?

The proposed algorithm shall compare with the other methodology within recently five years journal papers to prove the paper contributions.

It’s not only to derive the mathematical expressions for the proposed method system but also implemented by applications for getting more comparisons.

Author Response

Dear reviewer, thank you so much for the efficient review process and all the useful comments. Below are the manuscript’s revision descriptions and explanation of some unclear statement.

1) We have rephrase the description of the simulation part to make it more clear. The comparison is between different resource allocation schemes not between symmetric channel condition and asymmetric channel condition. Since algorithm 1 shows that the asymmetry of the channel affects the optimal parameters, and thus further affects the outage performance, in figure 1, we choose two channel conditions to assesses the system performance: symmetric channel condition (d1 = d2 = 5m) and asymmetric channel condition (d1, d2) = (3, 7)m. Note that (d1, d2) = (3, 7)m is just a case of the asymmetric channel condition which we choose to reflect the system performance. The effects of relay deployment (channel condition effects) is shown in figure 2.

2) The proposed algorithm is developed for the two way relay communication network which exploit power splitting energy harvesting protocol and decode-and-forward protocol to realize simultaneous wireless information and power transfer. So when encounter this kind of communication system, the proposed algorithm can be applied in the two way relay SWIPT communication network and lead the system obtain the best performance compared with other resource allocation schemes we mentioned in the simulation part.

3) As far as the author knows, there is no other methodology to jointly optimize power allocation and power splitting for the DF-TWR SWIPT network. And the innovation of our work is the designing of the joint resource allocation algorithm to minimize the system outage probability. So in order to verify the superiority of the proposed scheme, we have compared it with three typical traditional resource allocation schemes: Optimal \alpha Equal \rho, Optimal \rho Equal \alpha, Equal \alpha and \rho, and verify that the proposed scheme outperforms the benchmark schemes.

4) This manuscript aims at design a jointly resource allocation scheme to enhance the transmission performance of the communication system. We first formulate the system transmission model through mathematical expressions, and then formulate an optimization problem to jointly optimize power splitting ratio and power allocation ratio based on the transmission model. And in order to verify the superiority of the proposed scheme, we have compared it with three traditional resource allocation schemes. So it is a mathematical derivation based on communication application.

Thank you!

Round  2

Reviewer 1 Report

The authors have addressed all my concerns satisfactorily. Therefore, I recommend the paper for publication.